# Efficient V2V Communications by Clustering-Based Collaborative Caching †

**Hiroki Tokunaga and Suhua Tang \*** 

Graduate School of Informatics and Engineering, The University of Electro-Communications, Chofu, Tokyo 182-8585, Japan; h.tokunaga@can.lab.uec.ac.jp
\* Correspondence: shtang@uec.ac.jp; Tel.: +81-42-443-5308
† This paper is an extended version of our paper published in Tokunaga, H.; Tang, S. Efficient V2V communications by clustering-based collaborative caching. In Proceedings of the 2022 IEEE International Conference on Vehicular Electronics and Safety (ICVES), Bogota, Colombia, 14–16 November 2022; pp. 1–6, doi: 10.1109/ICVES56941.2022.9986927.

**Abstract:** Vehicle-to-vehicle (V2V) communication plays an important role in enabling autonomous driving. However, when multiple vehicles request the same content, like road conditions, delivering it individually by V2V communication can significantly increase traffic volume, potentially causing congestion in the wireless channel. To address this issue, Content-Centric Network (CCN) technology is applied to V2V communication, which improves communication efficiency by exploiting content cached at vehicles. However, previous methods faced the following challenges: (i) vehicles could not use content stored in nearby vehicles outside the communication path, and (ii) redundant caching of the same content occurred at nearby vehicles. To tackle these challenges, this paper proposes a collaborative caching method in which vehicles are grouped into clusters and each cluster has a designated head responsible for managing caches across all vehicles within the cluster. In this way, this method enables vehicles to use the content cached at adjacent vehicles that are not directly on a communication path. In addition, it eliminates redundant caches, allowing a more diverse range of content storage. Extensive simulation results demonstrate that the proposed approach effectively reduces content delivery latency by 33% compared to the method using clusters without cooperative caching and by 19% compared to the ECV+ method.

**Keywords:** V2V communication; Content-Centric Network; collaborative caching; clustering

## 1. Introduction

Vehicle-to-Vehicle (V2V) communication stands out as a pivotal technology in the field of autonomous driving [1]. The exchanged content can be categorized into two types: content related to safety and content related to comfort. The former typically has a brief lifespan and is seldom requested again. In comparison, the latter has a longer lifespan, and various vehicles may request the same information multiple times. When this type of content is delivered through separate pathways, the traffic volume increases rapidly with the growing number of requesting vehicles.

The Content-Centric Network (CCN) architecture [2] is specifically designed for the repeated delivery of identical content. In CCN, nodes use interest packets to request content and data packets to deliver it. Each node stores received or forwarded content as caches, enabling them to respond to requests instead of content providers. This approach effectively reduces communication delays and network traffic, making CCN widely adopted in wired networks. Recently, there has been a growing interest in applying CCN to vehicular networks [3–5], where a vehicle is treated as a CCN node.

There are limitations and issues with the basic CCN, however.

(1) This approach cannot use content cached at nearby nodes that are not on communication routes.

(2)   Nodes autonomously decide whether to cache content, leading to the possibility of multiple vehicles storing the same information. This consumes cache buffers uselessly and hampers the caching of other content.

(3)   The mobility of vehicles makes it complex to establish concrete communication paths.

Collaborative caching provides an effective solution to address (1) and (2). As for (1), Ref. [6] solves it by letting vehicles periodically broadcast beacons informing the names of cached content with great overhead due to the extensive communication cost. As for (2), Refs. [3,7,8] suggest that each vehicle make a probabilistic decision on whether to cache content. While this reduces the redundancy of storing identical content in nearby vehicles, it does not completely eliminate it.

To further improve the performance of vehicular networks by CCN, this paper proposes a novel collaborative caching method. This method groups vehicles into clusters and manages caches in each cluster unitarily. Specifically, each cluster comprises a cluster head (CH) and several cluster members (CM). A CH is responsible for dealing with content requests and managing content in its cluster. When a CH receives a content request, it searches for the requested content in its cluster, enabling the CH to respond to requests based on caches from any vehicles within its cluster. When a CH receives content, it determines whether to store it in its cluster and if it does, it also decides which vehicle in its cluster stores it, which prevents duplicated content from being stored in a cluster.

The contributions of this paper are two-fold as follows:

- To the best of our knowledge, this is the first work that proposes to consolidate content management within a cluster to optimize the utilization of cached content and cache buffers in V2V communications.
- We implement unitary cache management by letting each vehicle overhear data packets, which suppresses the control overhead.

Part of this paper has been presented in a conference [9]. In this work, we added the estimation of content popularity and on this basis refined cache management. In addition, we evaluated the proposed method in more scenarios: both the freeway scenario used in [9] and an urban model scenario that has been newly added in this paper. We also added a comparison with ECV+ [3]. Extensive simulation evaluations demonstrated that the proposed method effectively increased the cache hit rate within a cluster and improved network efficiency.

The rest of this paper is organized as follows. Section 2 reviews the basic CCN and related works. Section 3 presents the proposed method, and Section 4 confirms its effectiveness by simulation evaluation under different settings. Finally, Section 5 concludes this paper.

## 2. Related Work

Here, we introduce the basic CCN and review related work on clustering algorithms, collaborative caching methods, and routing protocols for vehicular networks. Table 1 shows a brief explanation of each related method.

**Table 1.** Comparison of related methods.

| Reference | Kind | Description |
|---|---|---|
| [10] | Cluster construction | Construct clusters based on inter-vehicle distances, vehicle heights, and route stability |
| [11] | Cluster construction | Clusters may be constructed with multi-hop distances |
| [3] | Reducing the cache redundancy | Cache probability is determined by channel usage |
| [7] | Reducing the cache redundancy | Cache probability is determined by content similarity and vehicle mobility |
| [8] | Reducing the cache redundancy | Cache probability is determined by the characteristics of vehicles' mobility |

**Table 1.** *Cont.*

| Reference | Kind | Description |
|---|---|---|
| [4] | Collaborative caching | CHs receive content from RSU; CMs receive content from both of them |
| [12] | Collaborative caching | Vehicles broadcast alternately |
| [5] | Collaborative caching | A vehicle preferentially requests content from others moving in the same lane |
| [13] | Collaborative caching | A vehicle decides whether to cache content based on the vehicle's power, the content's popularity, the gain acquired by the content, and the distance (hop count) to the content provider |
| [14] | Collaborative caching | Both vehicles and RSUs construct clusters |
| [15] | Collaborative caching | Vehicles within an RSU are divided into clusters, and they send requests to the RSU |
| [16] | Collaborative caching | Vehicles construct clusters and receive requested content from either their CHs or RSUs; part of the mathematical model is solvable as a knapsack problem |
| [6] | Collaborative caching | Vehicles broadcast the names of cached content |
| [17] | Collaborative caching | Cache policy is determined with deep reinforcement learning |
| [18] | Collaborative caching | The problem of which content to store and where to store it is determined by machine learning |
| [19] | Collaborative caching | Data carrier node is selected by reinforcement learning |
| [20] | Cache place determination | Cache placement is treated as an MWVCP problem |
| [21] | Cache place determination | Node $n$ ($n < k$) only stores content $c$, where $c$ mod $k = n$ |
| [22] | Relay vehicle determination | Vehicles that lie in the common communication area are preferentially selected |

### 2.1. Basic CCN [2]

When a node requests content in CCN, it sends an interest packet containing the content name towards the content provider. If a node receiving the interest packet has the requested content, it replies with a data packet containing the content. Otherwise, it records the content name and the ID of the requesting node in its Pending Information Table (PIT) and forwards the interest packet towards the content provider according to its Forwarding Information Base (FIB). When a node receives a data packet, it stores the content included in the packet in the Content Store (CS) and forwards the packet towards the requesting node according to its PIT. Generally, CCN was designed for wired networks and has no node mobility considerations.

### 2.2. Cluster Construction

Clustering is a typical method of node management in vehicular networks and thus is a target of research [10,11]. These methods, however, are designed for general vehicular communication and are not optimized for CCN.

### 2.3. Collaborative Caching

Ref. [4] studies a combination of CCN and clustering whereby a CH stores content received from a Road-Side Unit (RSU), making it possible to reply to requests from CMs even if they are outside of the transmission ranges of RSUs. However, this method does not use cache buffers at CMs at all.

In the cooperative scheduling method [12], each RSU on a road divides vehicles in its transmission range into clusters in a centralized way. In each cluster, each vehicle takes turns to broadcast content according to the specified schedule. Although this helps to

improve channel efficiency, it is not applicable to the scenario wherein vehicles request different content.

Refs. [3,7,8] studied how to reduce the redundancy in content caching by storing received content with a probability. These methods help to reduce the chance that neighbor vehicles save duplicated content, but they cannot prevent it entirely.

In [13], vehicles decide whether to cache content based on the vehicle's power, content popularity, the gain acquired by caching the content, and the hop count of the content.

In [14], both vehicles and RSUs form clouds: a vehicular cloud (VC) and an RSU cloud (RC), respectively. When a vehicle requests content, it first asks its VC. If all vehicles in the cloud do not have the content, it queries its associated RSU. The RSU replies if one of its RCs has the content or fetches the content from the remote server otherwise.

Some research has applied existing algorithms to V2V communications, e.g., a cuckoo search for route calculation [15], the 0–1 knapsack problem for cache decisions [16], and the Minimum Weight Vertex Cover Problem for cache placement [20].

With the emergence of AI, some studies suggest combining AI and vehicular networks: Ref. [17] calculates the cache policy with deep reinforcement learning, Ref. [18] uses machine learning to decide which content to store and where to store it, and Ref. [19] uses reinforcement learning to decide which vehicle would be the best data carrier node.

To share content information, Ref. [6] suggests letting each vehicle broadcast the names of its cached content to neighbor vehicles by using periodical beacon messages. Neighbor vehicles, on receiving this message, store the information and use it to check which vehicle stores content as a cache. However, the notification of content information via beacons causes huge overhead and increases the possibility of packet collision.

Ref. [21] targets a stationary environment wherein each node requests a time-shifted TV program. Each content router (CR) has a label $n$ of a positive integer, where $n < k$. CR $n$ stores only chunks $c$, where $c \mod k = n$, which prevents multiple nodes from caching the same content. It is difficult, however, to apply this method in the V2V environment where nodes move.

### 2.4. Determining Request/Relay Vehicle

Some research has discussed how to select a vehicle to request or relay content. In [22], a vehicle that lies in the common communication area of a packet sender and other potential relay vehicles is preferentially selected. In [5], the content requester requests content from vehicles moving in the same lane first and then to vehicles moving in adjacent lanes upon failure.

### 2.5. Comparison with Related Work

The purpose of this paper is not to construct clusters but to make cache management more efficient by using the clusters. Each CH manages content in its cluster unitarily, which solves the problem of cache duplication that was a challenge in previous works [3,7]. In this way, vehicles can cache more content in a cluster, which enables CHs to reply to diverse requests locally instead of further fetching them from other vehicles or content providers.

## 3. The Proposed Method

Figure 1 shows the overview of the proposed method. It mainly consists of four components as follows:

- Cluster construction: Vehicles form clusters autonomously based on their moving speeds and directions. Each cluster has a cluster head (CH) and a variable number of cluster members (CMs).
- Content fetching: On requesting content, each CM sends an interest packet to its associated CH. Each CH is responsible for forwarding interest and data packets and replies to an interest packet if the requested content is found in its cluster.
- Popularity management: Vehicles update the popularity of the content by monitoring requests from its cluster and the interest packets it forwards.

- Cache management: Each CH manages cache buffers for itself and all its CMs. On receiving a data packet, it decides whether and where to store the content. If there is a shortage of cache buffers, the content with the lowest popularity is replaced by the new content.

With these steps, the proposed method removes duplicated caches and enables each cluster to cache more content locally. On this basis, it reduces the number of packet transmissions and the response time accordingly.

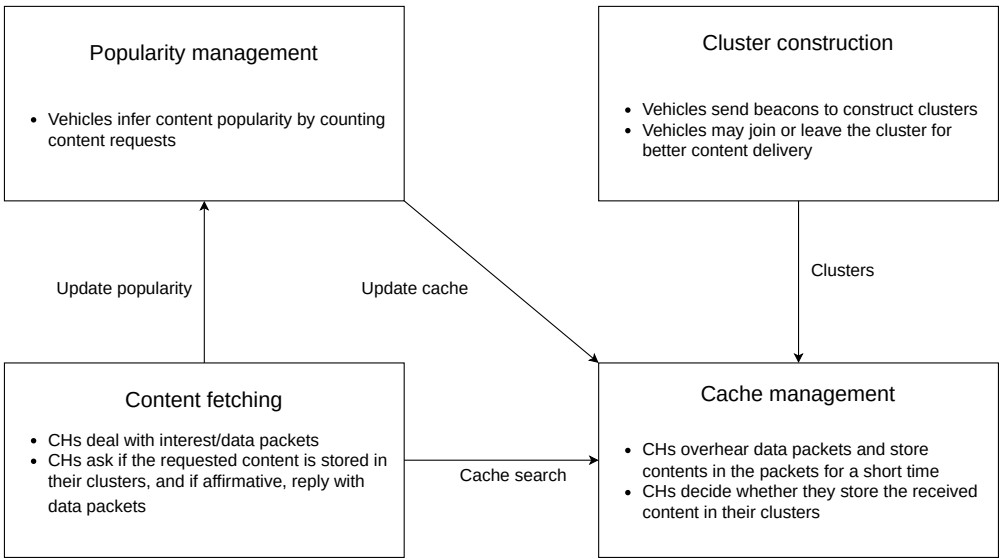

**Figure 1.** The overview of the proposed method.

*3.1. Cluster Construction and Maintenance*

3.1.1. Detecting Neighbors by Exchanging Beacons

Each vehicle periodically broadcasts a beacon to alert nearby vehicles of its presence. Before broadcasting a beacon, the sender removes from its neighbor list all neighbor vehicles for which their information has not been updated within $t_{\text{neighbor}}$. Each beacon contains the senders' vehicle ID, position, speed, the number of neighbor vehicles, a flag indicating whether the sender is a CH, the ID of the associated CH if the sender is a CM, and the connection expired predicted (CEP) CH ID if the sender vehicle is a CH. When a vehicle receives a beacon, it records the sender as its neighbor vehicle.

3.1.2. CM Joining/Leaving

Every vehicle is equipped with a CM table designed to handle CM information in case it becomes a CH later. The CM table comprises two essential components for each CM entry: (i) a list of content with names and timestamps that denote the reception time and (ii) a cluster-keep-alive timer. This timer is refreshed upon receiving beacons from the CM and triggers the removal of the CM entry if it expires.

Each vehicle initially assumes the role of a CH. A CH without any CMs in its cluster is identified as an Orphan Vehicle (OV). When an OV receives a beacon from another CH, it transmits a beacon claiming its joining into the CH's cluster as a CM if specific conditions are satisfied as follows:

- Both the beacon sender and receiver move in the same direction.
- The distance between the two vehicles is less than or equal to $d_{cluster}$.
- The relative velocity between the two vehicles is less than or equal to $v_{cluster}$.

When a CH receives a beacon from a CM for the first time, it adds the CM's information to its CM table. When the CH receives a beacon from the CM again, it resets the CM's cluster-keep-alive timer. When a CM receives a beacon from its CH, it resets its cluster-keep-alive timer.

When one of the following conditions is satisfied, a CM leaves its cluster, and its CH considers that the CM has left its cluster.

- The cluster-keep-alive timer expires at either a CM or a CH.
- The distance between a CH and a CM is more than $d_{cluster}$.

### 3.1.3. Splitting/Merging Clusters

If the distance between two adjacent CHs changes drastically, it is necessary to split a cluster into two (when the distance is too large) to avoid relay failure or to merge two clusters (when the distance is too short) to improve relay efficiency.

To this end, a CEP CH ID is added to the beacons sent from CHs. A CEP CH is a neighbor CH of the sender CH that satisfies the following conditions.

- The CEP CH, as a relay of the sender CH, has the shortest distance to the CH in either the front or the back of the CH.
- The connection between the CEP CH and the sender CH is predicted to break within $t_{break}$.

A beacon may contain at most two CEP CH IDs to notify the potential relay failure of the sender CH briefly. Figure 2 shows an example, where CH1, predicting that the connection between itself and CH2 will break soon, sends a beacon containing CH2's ID as a CEP CH ID.

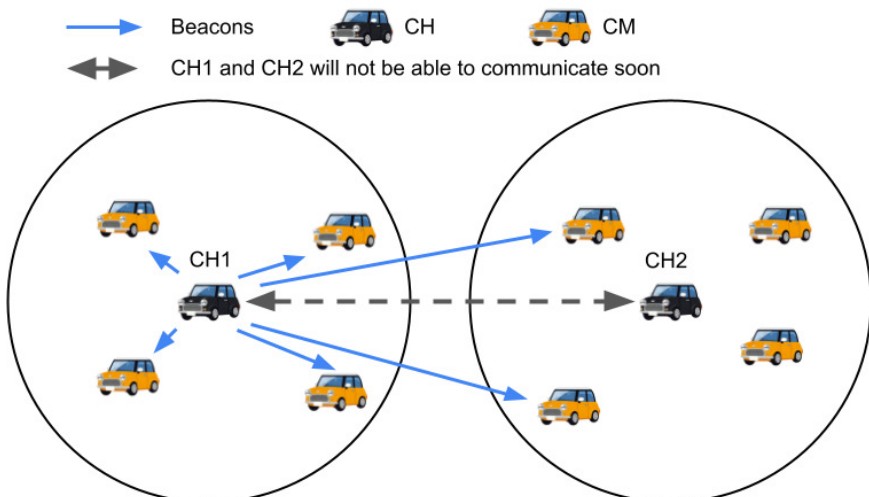

**Figure 2.** CH sends a beacon containing a connection-expired-predicted CH ID.

When a CM receives a beacon from any CH containing a CEP CH ID and it can communicate with both the sender CH and the CEP CH, it tries to leave its cluster immediately and create a new one. Because multiple CMs simultaneously receive a beacon with a CEP CH ID, each CM calculates its CH suitability according to (1). Here, $N_i$ denotes the number of neighbors of vehicle $i$, and $\bar{d}_i$ is the average distance from vehicle $i$ to its neighbors.

$$s_i = \frac{N_i}{\bar{d}_i}. \tag{1}$$

Then, the vehicle with the largest $s$ becomes a new CH, and other vehicles choose to become its CMs or remain as the CMs of the previous clusters. Figure 3 shows that CH3 constructs a new cluster based on information in a beacon following the actions in Figure 2.

Two clusters start to merge when all of the following conditions are satisfied.

- The CHs of both clusters are moving in the same direction.
- A CH can communicate with all of the CHs that the other can.
- The distance between the two CHs is less than or equal to $d_{cluster}$.
- The CH has not sent a merge request to another CH.

When CH1 decides to merge its cluster with CH2's, it sends a merge request to CH2 and schedules a merge-cancel timer. CH2 accepts the request if the following conditions are satisfied and ignores the request otherwise.

- The cluster merging requirements are satisfied.
- CH2 is not requesting to merge clusters with another CH.

Once CH2 accepts the request, it selects a new CH from CH1 and CH2 by calculating both of the CH's CH suitability according to (1) and choosing the CH with higher suitability. Then, it sends CH1 a merge ACK packet containing the information on whether CH2 becomes a CH or CM. All CMs are notified of the cluster merging by CH's beacon or the merge ACK packet.

Cluster merging is canceled if the merge-cancel timer expires.

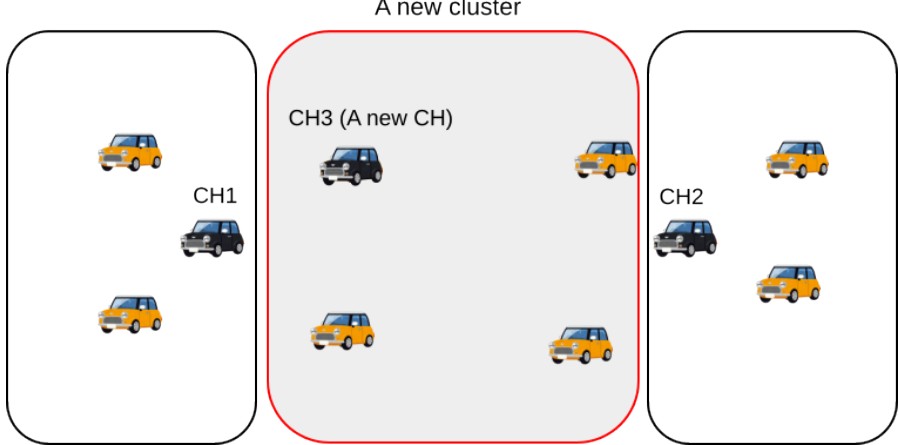

**Figure 3.** After receiving the beacon, a new cluster will be created.

### 3.2. Content Fetching

Here, we explain how content is fetched in the proposed method using Figure 4.

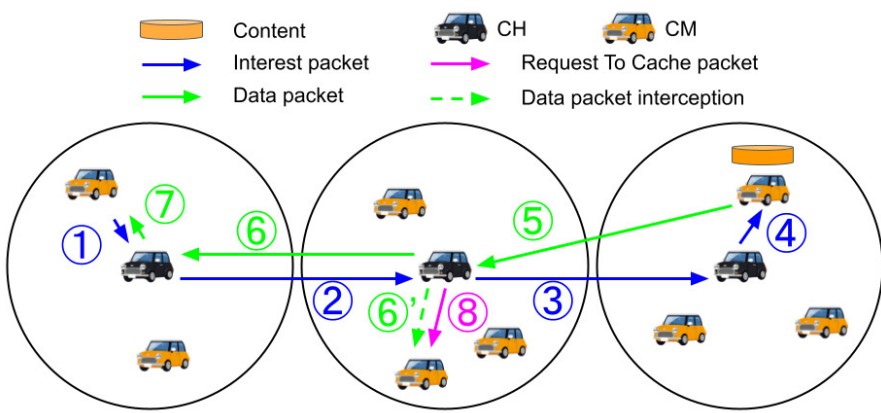

**Figure 4.** Communication process of the proposed method.

When a CM requests content, it sends an interest packet containing the name of the requested content to its CH (step 1 in the figure). The CH tries to reply to the request in the following order.

- If the CH itself has the requested content, it replies with a data packet.
- If one of its CMs has the content, the CH adds the ID of the request vehicle to the interest packet and forwards it to the CM that has the cached content (step 4 in the figure).
- If no vehicle in its cluster has the requested content, the CH first tries to forward the interest packet to other CHs according to its FIB (steps 2 and 3 in the figure).

When a CM receives an interest packet from its CH and has the requested content, it sends a data packet to the requester (step 5 in the figure). The CM communicates with the requester directly if it can, and otherwise, it relays the content via its CH. Due to potential packet loss in the wireless environment, a CM may not have the requested content. In such cases, the CM sends a content-not-found message to its CH. Then, the CH, accordingly, forwards the interest packet towards the content provider via other CHs.

A CH, on receiving a data packet, forwards it to the content requester according to the PIT (steps 6 and 7 in the figure).

In some cases, there may be no CH available as a forwarder. A CH forwards an interest packet to the CM that is closest to the content provider, using this CM as a relay. When a CM receives an interest packet from a CH outside of its cluster, it sends the interest packet to its CH which further forwards this towards the content provider.

Algorithm 1 is the pseudo-code for interest and packet handling.

---

**Algorithm 1:** Packet management

```
 1  Procedure OnReceivingInterestPacket
 2      if Am I a CH? then
 3          if Do I store the requested content? then
 4              │  Reply with a data packet ;
 5          else if Does one of the CMs store the requested content? then
 6              │  Forward the interest packet to the CM ;
 7          else
 8              │  Forward the interest packet towards the content provider ;
 9          end
10      else
            // Interest packet handling for a CM
11          if Do I store the requested content? then
12              if Is it possible to send a packet to a one-hop-before vehicle? then
13                  │  Send a data packet to the vehicle ;
14              else
15                  │  Send a data packet to the CH ;
16              end
17          else
18              │  Send a content-not-found packet to the CH ;
19          end
20      end
21  end
22  Procedure OnReceivingDataPacket
23      if Am I a CH? then
24          if Is the packet destination one of my CMs? then
25              Forward the packet to the CM ;
26              if Does the CM have empty buffers? then
27                  │  Record that the CM has the content in the buffer ;
28              else
                    // StoreContentInCluster is defined later.
29                  │  Call StoreContentInCluster with the content as the parameter ;
30              end
31          else
32              Forward the packet according to PIT ;
33              Call StoreContentInCluster with the content as the parameter ;
34          end
35      else
            // Data packet handling for a CM
36          if Does my buffer have empty space then
37              │  Store the content in the buffer ;
38          else
39              Consume the content immediately ;
40              Store the content in the temporary buffer ;
41          end
42      end
43  end
```

---

### 3.3. Content Popularity Inference

Each vehicle has a request counter table (Figure 5) to count how many times a content is requested to infer the content's popularity. It involves both requests from inside the cluster and the requests from other clusters that pass this cluster. Interest packets and data packets contain a timestamp. This timestamp, together with the content name, serves as a unique Request ID. When a vehicle receives one of these packets, it checks whether the

Request ID in this packet is fresher than the one in the corresponding entry of the table, and if affirmative, it increases the counter by 1 and updates the Request ID. In this way, the counter of the content reflects its popularity.

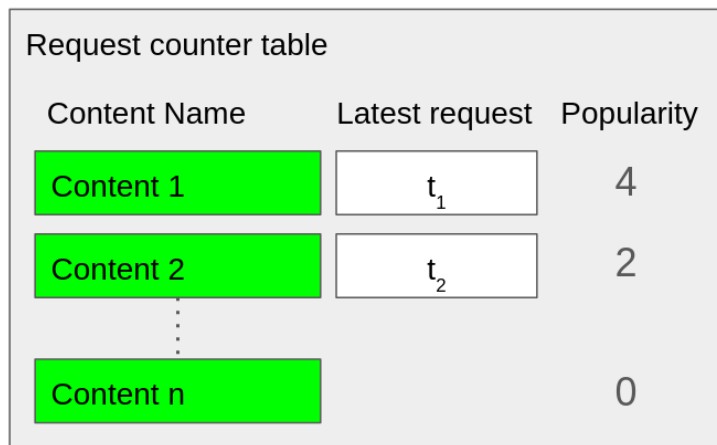

**Figure 5.** Request counter table.

Algorithm 2 is the pseudo-code for updating content's popularity.

---

**Algorithm 2:** Updating content's popularity

---

1 **Procedure** *UpdateContentPopularity*
   **Data:** Input: *contentName*, *requestTimestamp*
2   **if** *Does the vehicle's request counter table have the entry for contentName?* **then**
3     *timestampInTable* ← the timestamp recorded in the table ;
4     **if** *Is requestTimestamp newer than timestampInTable?* **then**
5       Increase the popularity of *contentName* by 1 ;
6       Update the timestamp in the table entry with *requestTimestamp* ;
7     **end**
8   **else**
9     Create an entry for *contentName* with its request timestamp
      *requestTimestamp* ;
10   **end**
11 **end**

---

### 3.4. Cache Management

A CH manages how to cache content in its cluster. When a CH receives or forwards a data packet, it uses the following operation.

- If the packet destination is its CM and the CM has free cache space, the CH adds the timestamp and the name of the content to the corresponding CM's entry in its CM table.
- Otherwise, the CH decides whether to save the content or not by checking if the content is already stored in the cluster. If it saves the content, it also decides which vehicle in the cluster to save it to. If the target vehicle is a CM, it sends a request-to-cache message to the CM and adds the timestamp and the name of the content to the entry of the corresponding CM in its CM table (step 8 in Figure 4). If the buffer of the CM has no free space, the CM decides which content to replace based on content popularity. Figure 6 shows this operation.

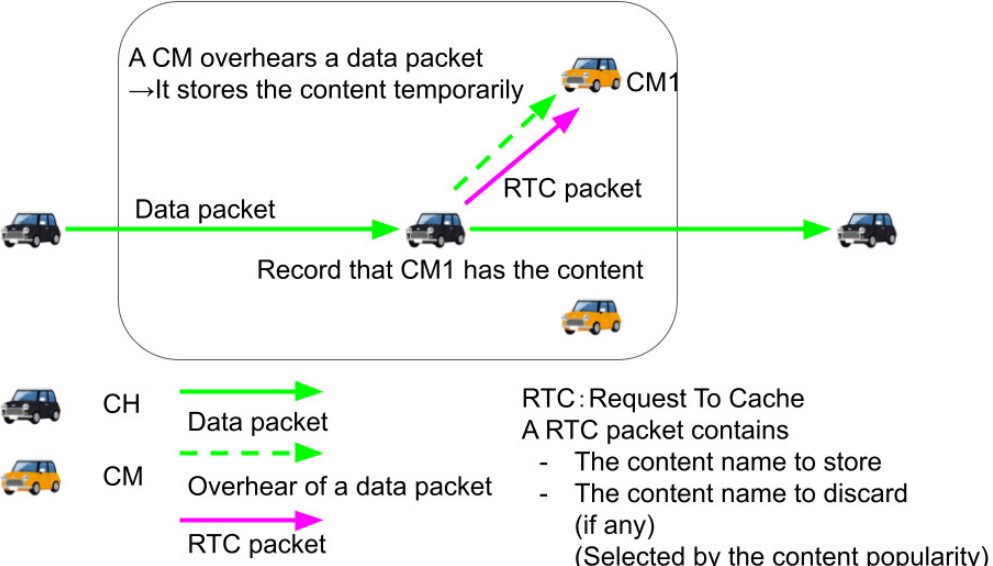

**Figure 6.** Data packet forwarding operation.

A CM overhears data packets and stores content in its temporary buffer (step 6' in Figure 4). When a CM receives a request-to-cache message from its CH, it moves the specified content from its temporary buffer to its CS (Content Store). If a specified time has passed and the CM has not received a request-to-cache message, it discards overheard content silently.

When a CH selects a CM to store content, it selects the vehicle with the maximum free space. If no CMs have free cache space, the CH selects a CM with content with the least popularity. Then, the selected vehicle replaces the content with the new content using the same cache policy as the CH.

When a CM joins a cluster, it sends a sync message that contains information about the $n_{\text{sync}}$ most popular content data (with content names and their timestamps) in its buffer to its CH. The CH updates its CM table accordingly. When a CH considers that a CM has left its cluster (cluster-keep-alive timer expires), it removes the CM's information from its CM table.

Algorithm 3 is the pseudo-code for storing content in a cluster.

---

**Algorithm 3:** Storing content in a cluster

1 **Procedure** *StoreContentInCluster*
    `// Assumption:  The caller is a CH`
    **Data:** Input: *content*
2   **if** *Do I have empty buffers?* **then**
3     Store the content ;
4   **else if** *Does one of the CMs have empty buffers?* **then**
5     Send a request-to-cache packet to the CM ;
6   **else**
7     $n \leftarrow$ ID of the vehicle in the cluster having the least popular content ;
8     **if** *Is n the ID of the caller?* **then**
9       Store the content ;
10     **else**
11       Send a request-to-cache packet to the vehicle with ID *n* ;
12     **end**
13   **end**
14 **end**

---

## 4. Simulation Evaluation

We evaluated the proposed method with clustering, collaborative caching, and content popularity inference (C-CoCach-Pop) using Scenargie [23]: a commercial simulator supporting V2V communications.

### 4.1. Comparison Methods and Evaluation Metrics

We compared the proposed method with the following three methods. 'Pop' in a method name indicates that popularity inference is used in the method.

- V2V communication using basic CCN [2] without clustering (BCCN-Pop): A vehicle on a communication path caches the content on receiving a data packet, while the cache buffer for a vehicle off the path is not exploited.
- V2V communication with clustering enabled but collaborative caching disabled (C-BCCN-Pop): Here, clusters are constructed for forwarding packets in the same way as in the proposed method.
- V2V communication with clustering and collaborative caching but without content popularity inference (C-CoCach [9]): In this method, cache management is based on the Least Recently Used (LRU) policy.
- ECV+ method [3] in which each vehicle decides the cache probability based on channel usage.

A vehicle sends a beacon periodically in both the proposed and comparison methods. A beacon only contains the sender's position and velocity in BCCN-Pop, while it includes the information for constructing and managing clusters in C-BCCN-Pop, C-CoCach, and C-CoCach-Pop. The simulation also evaluates the impact of overhead for managing CMs' caches.

We measured the median latency of data packets, channel usage, the success ratio of receiving requested content, and the cache hit ratio as metrics to evaluate the performance improvement of the proposed method.

### 4.2. Simulation Condition

We assume that a vehicle requests road condition information and weather information. The provider of content is the vehicle closest to it. Vehicles form clusters in the first 10 s, during which they do not request content. After that, each vehicle sends an interest packet periodically to fetch content. The content requests follow the Zipf law [24], where the number of requests of the $k$-th most popular content is proportional to $1/k$.

Detailed simulation conditions, simulation scenarios, the values of parameters, and the sizes of interest/data packets and control messages are shown in Table 2, Table 3, Table 4, and Table 5, respectively. Figure 7 is the open street map used in Scenario 3, which simulates real roads and buildings around Chofu Station.

We chose the values of parameters by initial experiments. Figures 8 and 9 show the median latency and the successful reception ratio to fetch content, respectively. From these results, we chose $d_{\text{cluster}} = 150$ m because with this value the successful reception ratio is high while the median latency is low.

**Table 2.** Simulation conditions.

| Key | Value |
| --- | --- |
| Simulator | Scenargie [23] |
| MAC protocol | IEEE 802.11p |
| Velocity of vehicles | 20 m/s |
| Vehicle transmission range | 350 m |
| Beacon transmission interval | 100 ms |
| Buffer size (number of content) | 0, 10, . . . , 100 |
| The interval for sending interest packets | 1 s |
| The lifetime of content in temporary buffer | 500 ms |
| Simulation time | 320 s |
| Number of trials | 8 |

**Table 3.** Simulation scenarios.

| No. | Road Shape | Number of Vehicles |
|-----|-----------|--------------------|
| Scenario 1 | 4 km straight road | 100 |
| Scenario 2 | 4 km straight road | 200 |
| Scenario 3 | 1.6 km × 1.6 km urban area | 300 |

**Table 4.** Values of parameters.

| Variable Name | Value |
|---------------|-------|
| $t_{\text{neighbor}}$ | 300 ms |
| $d_{\text{cluster}}$ | 150 m |
| $v_{\text{cluster}}$ | 1.4 m/s |
| $n_{\text{sync}}$ | 100 |
| $t_{\text{break}}$ | 1 s |

**Table 5.** Sizes of interest/data packets and control messages.

| Packet Name | Size (Bytes) |
|-------------|-------------:|
| Interest packet | 128 |
| Data packet | 512 |
| Content-Not-Found | 128 |
| Request-To-Cache | 128 |
| Sync (for $n$ content) | $24 + 12n$ |
| Beacon (clustering disabled) | 48 |
| Beacon (clustering enabled) | 80 |
| Merge request | 16 |
| Merge ack | 20 |

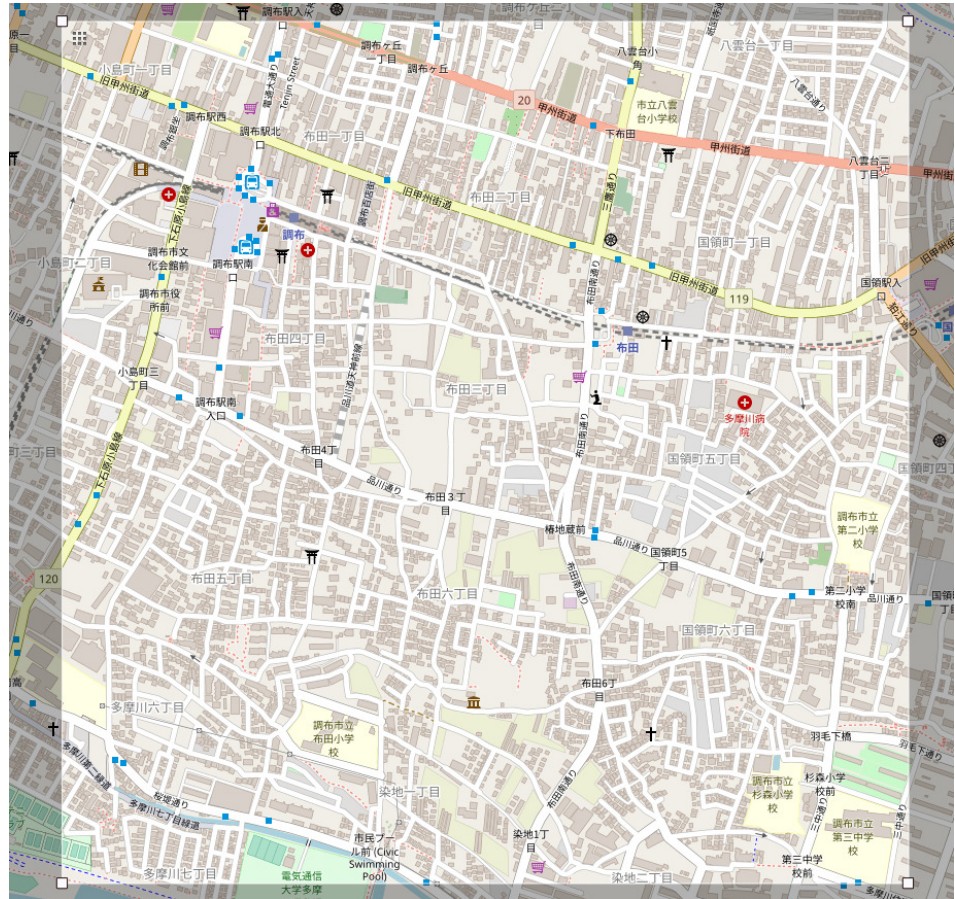

**Figure 7.** The map used in Scenario 3 (around Chofu Station in Japan) (©OpenStreetMap contributors).

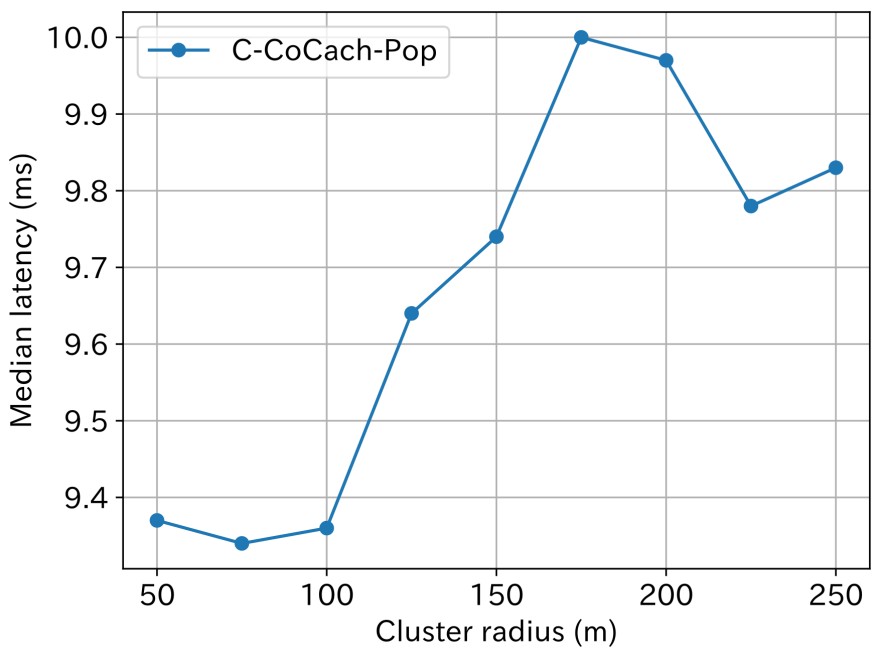

**Figure 8.** Median latency for retrieving the requested content with respect to cluster radius (Scenario 2).

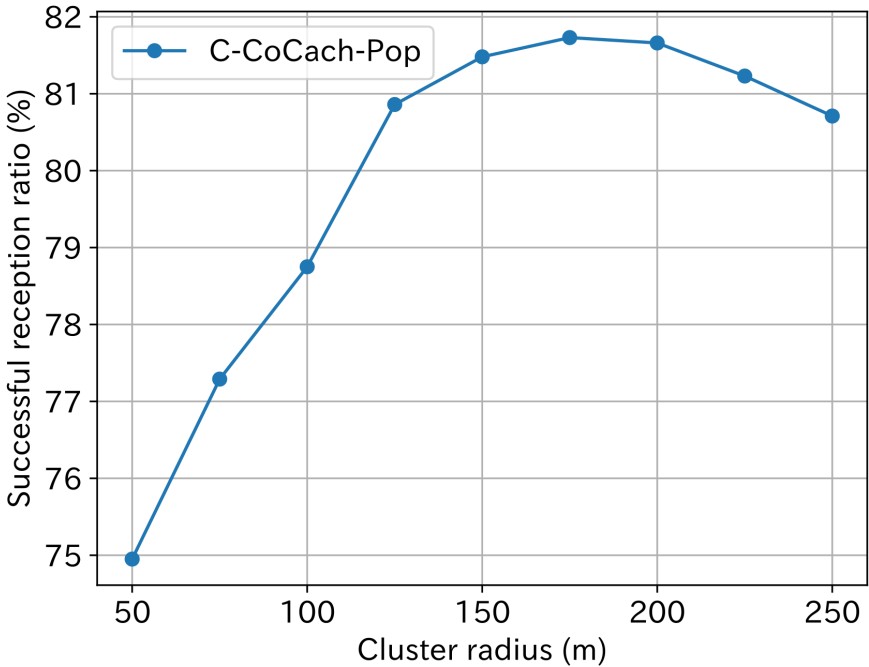

**Figure 9.** Success ratio of retrieving requested content with respect to cluster radius (Scenario 2).

*4.3. Simulation Results*

First, we evaluated the impact of popularity in the proposed method. Figures 10–13 compare C-CoCach without popularity inference and C-CoCach-Pop with popularity inference using Scenario 2 with 200 vehicles. By inferring content popularity and storing content with high popularity in a cluster, the in-cluster cache hit ratio is improved (Figure 13), and thus, the median latency for retrieving content is decreased (Figure 10). By shortening content delivery paths, the number of failures in content delivery is also decreased (Figure 11). Note that the overall cache hit ratio of C-CoCach-Pop is less than that of C-CoCach (Figure 12). The cache policy of C-CoCach is LRU, for which content popularity is irrelevant. Hence, vehicles store

both popular and unpopular content in C-CoCach, which increases content variety, resulting in a higher cache-hit ratio. But this increases the probability that some popular content is replaced by non-popular content that must be fetched again later.

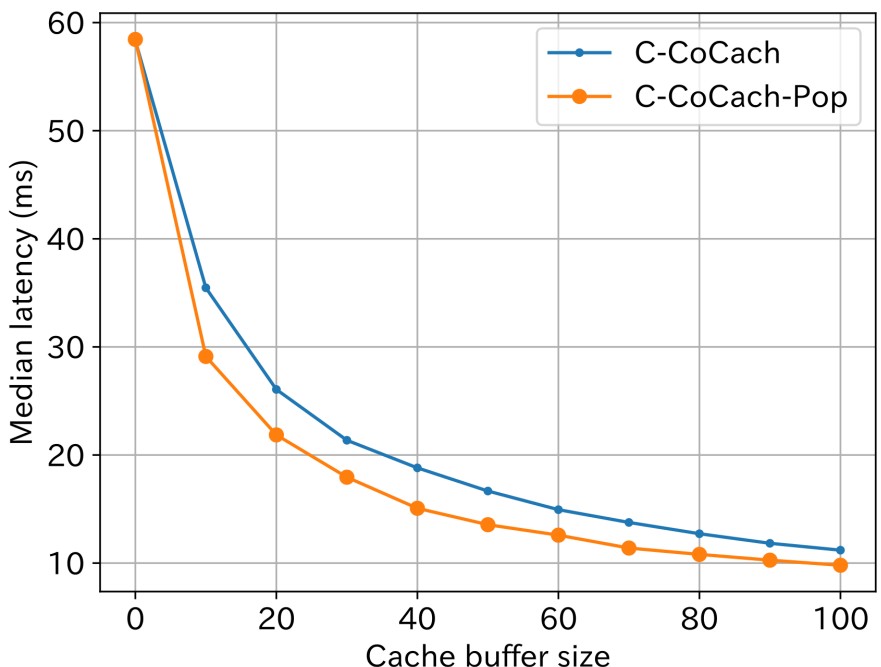

**Figure 10.** Median latency for retrieving the requested content with respect to cache buffer size (Scenario 2).

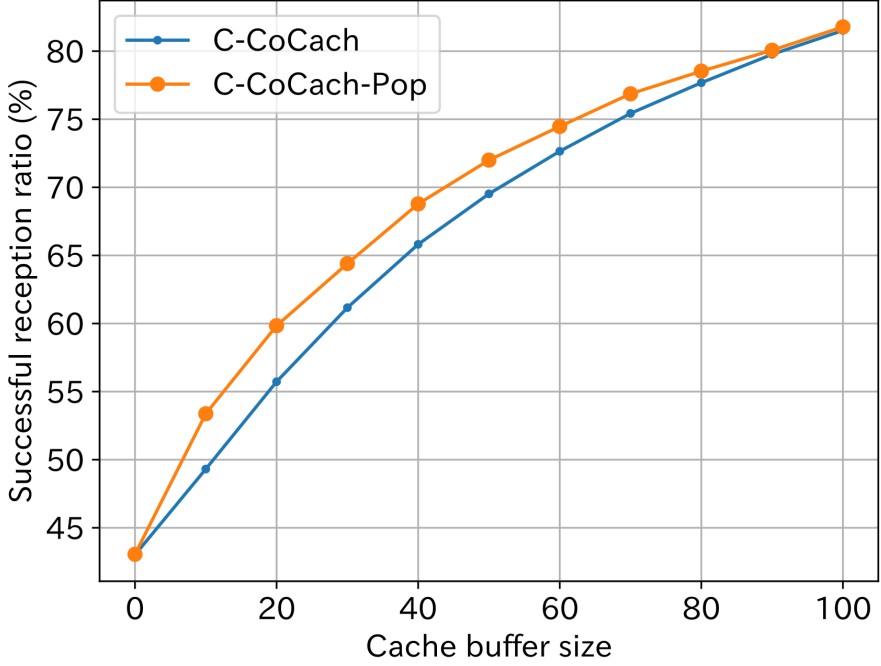

**Figure 11.** Success ratio of retrieving requested content with respect to cache buffer size (Scenario 2).

These results confirm the effect of using popularity inference in the proposed method. In the following, we consider that all methods use popularity inference and evaluate their performance using Scenario 1 at first.

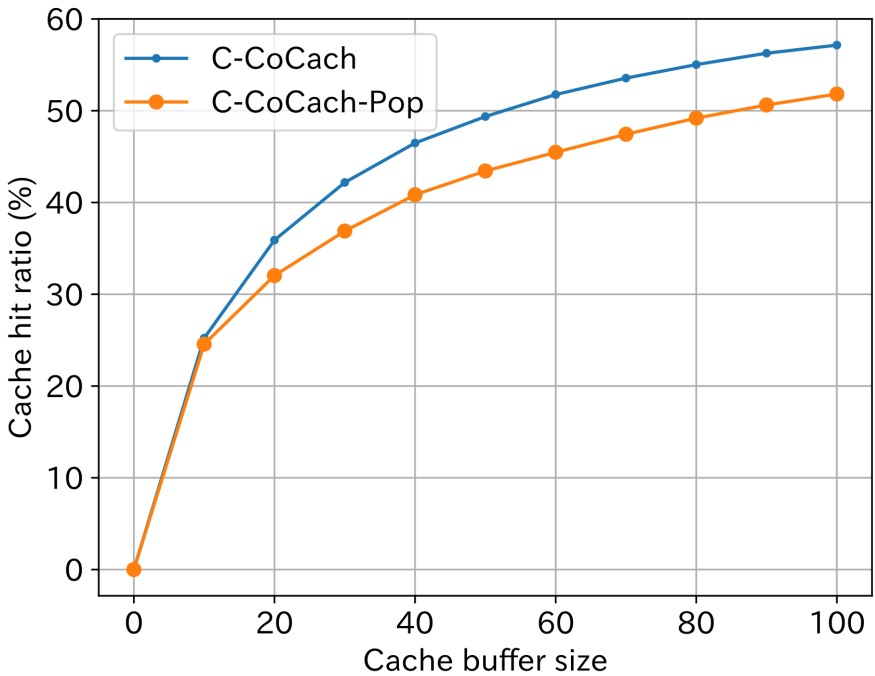

**Figure 12.** Cache hit ratio with respect to cache buffer size (Scenario 2).

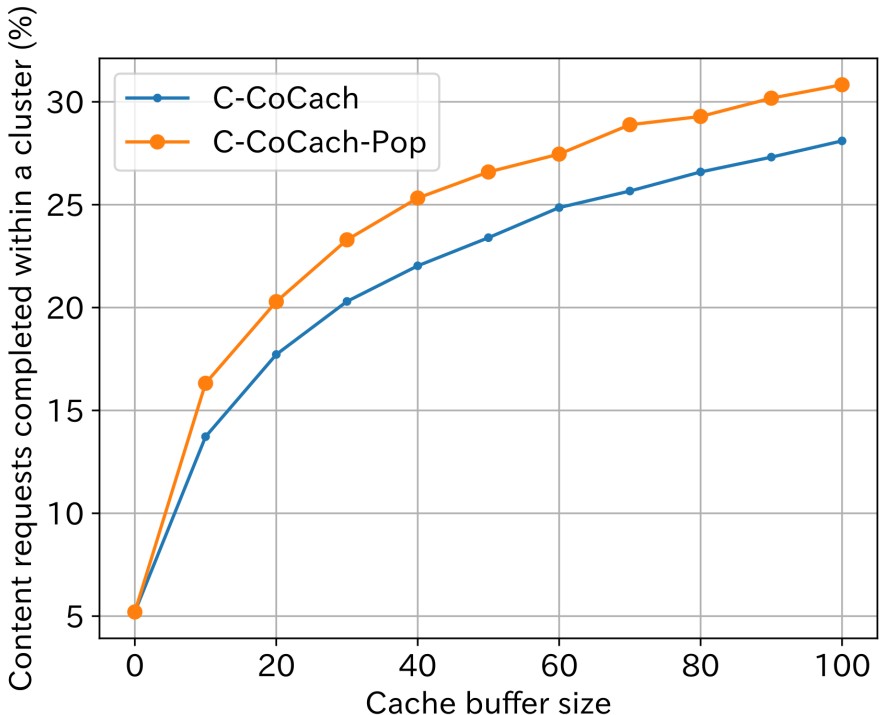

**Figure 13.** In-cluster cache hit ratio with respect to cache buffer size (Scenario 2).

Figures 14 and 15 show the median latency and the successful content reception ratio with respect to cache buffer size, respectively. Generally, a large cache buffer size enables the caching of more content, leading to small latency and a high success ratio in all methods. Compared with BCCN-Pop, the latency may be degraded a little in C-CoCach-Pop because almost all content goes through the CH, which increases the hop count. Compared with C-BCCN-Pop, which also uses clusters, the proposed C-CoCach-Pop method reduces the latency and increases the successful reception ratio in all ranges. This is because, with collaborative caching, more content can be fetched from local cache buffers, which reduces the hop count and also increases the successful reception ratio, as illustrated in Figure 15.

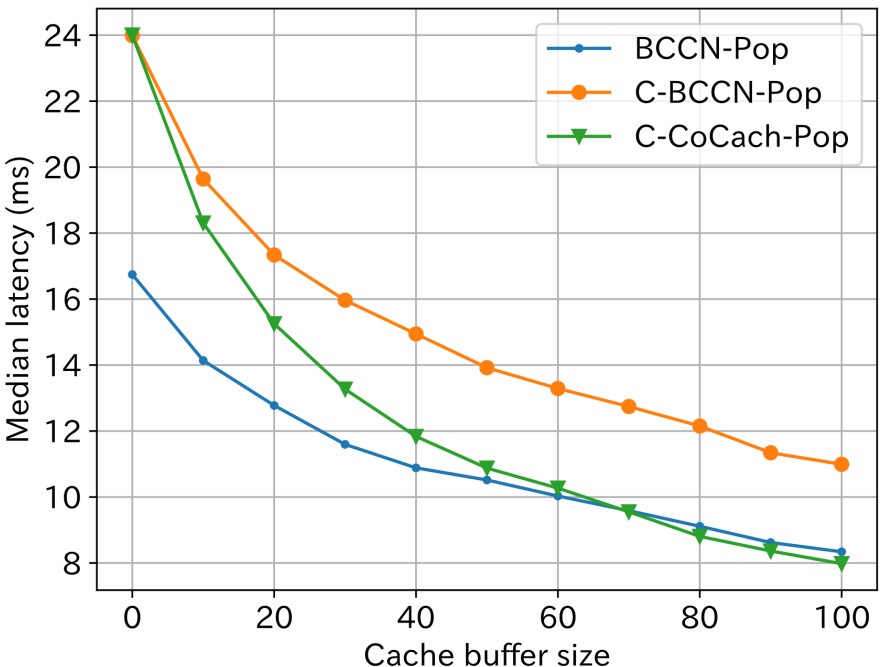

**Figure 14.** Median latency for retrieving the requested content with respect to cache buffer size (Scenario 1).

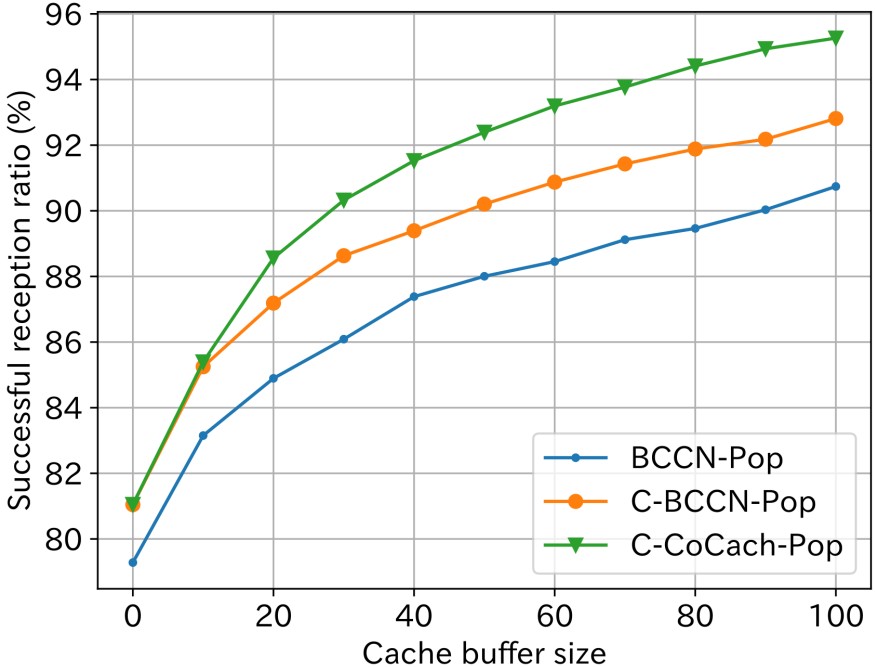

**Figure 15.** Success ratio of retrieving requested content with respect to cache buffer size (Scenario 1).

Figures 16 and 17 show the cache hit ratio and in-cluster cache hit ratio, respectively, with respect to the cache buffer size. Both increase with the cache buffer size. The in-cluster cache hit ratio indicates the percentage of content that is retrieved in one hop within a cluster, and it is effectively improved by C-CoCach-Pop. This is also confirmed in Figure 18, which illustrates the distribution of hop counts required to retrieve content.

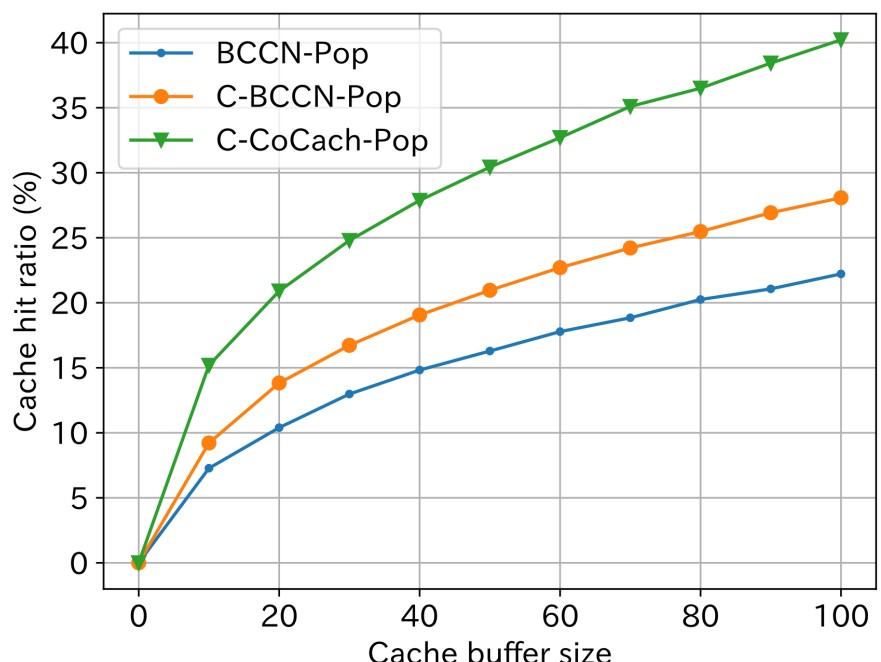

**Figure 16.** Cache hit ratio with respect to cache buffer size (Scenario 1).

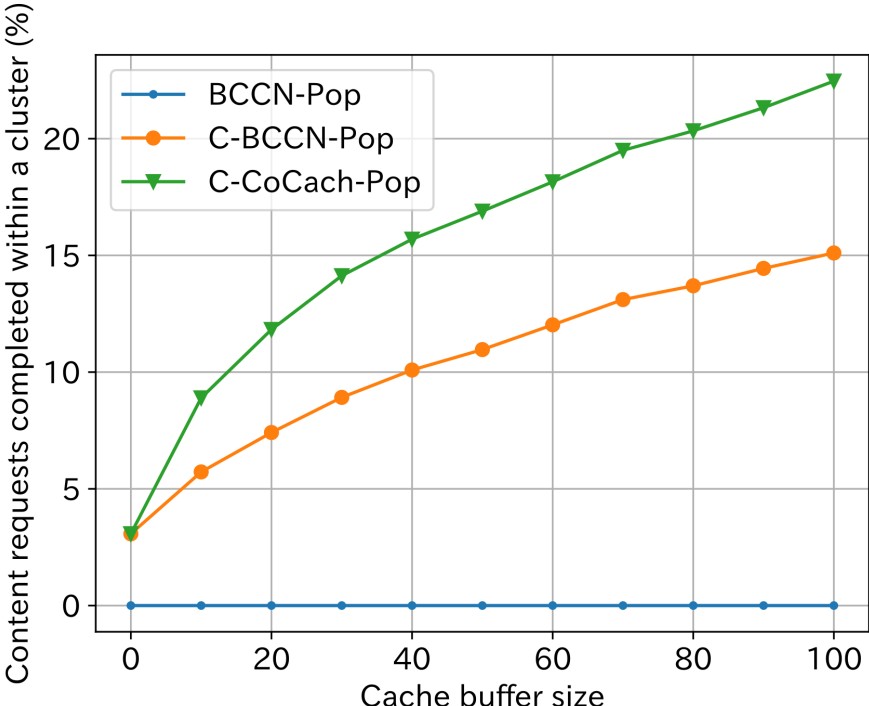

**Figure 17.** In-cluster cache hit ratio with respect to cache buffer size (Scenario 1).

Figure 19 shows the average channel usage when the buffer size is 100. BCCN-Pop has the lowest channel usage, and implementing collaborative caching increases the overhead. When considering the methods using clustering, C-CoCach-Pop may cause more overhead in cache management. But actually, the results show that C-CoCach-Pop has less channel usage than C-BCCN-Pop, which indicates that the reduction in channel usage by collaborative caching is greater than that caused by the management overhead.

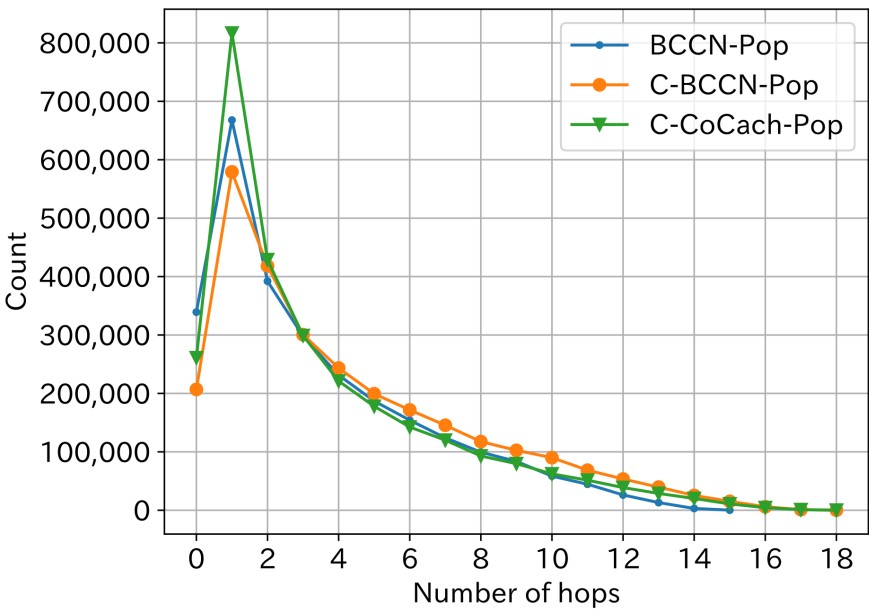

**Figure 18.** Distribution of hop count required to retrieve the requested content (Scenario 1, buffer size: 100).

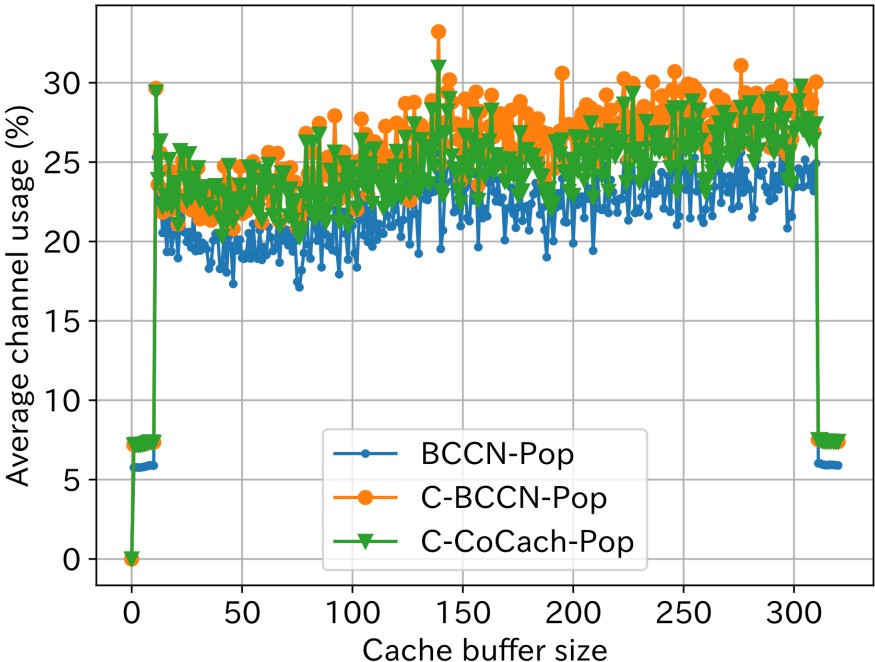

**Figure 19.** Average channel usage (Scenario 1, buffer size: 100).

Next, we investigated the impact of vehicle density. Figures 20 and 21 show the results for Scenario 2, for which the number of vehicles increases from 100 to 200, and the channel becomes more congested than in Scenario 1. Compared with Figure 14, the cache hit ratio in C-CoCach-Pop is improved. This is because clusters tend to contain more vehicles, and thus, more content can be cached in a cluster by exploiting the increased cache buffer. However, the successful reception ratio is degraded in all methods due to channel congestion. Compared with C-BCCN-Pop, our proposed method C-CoCach-Pop with collaborative caching still reduces the latency and improves the success ratio.

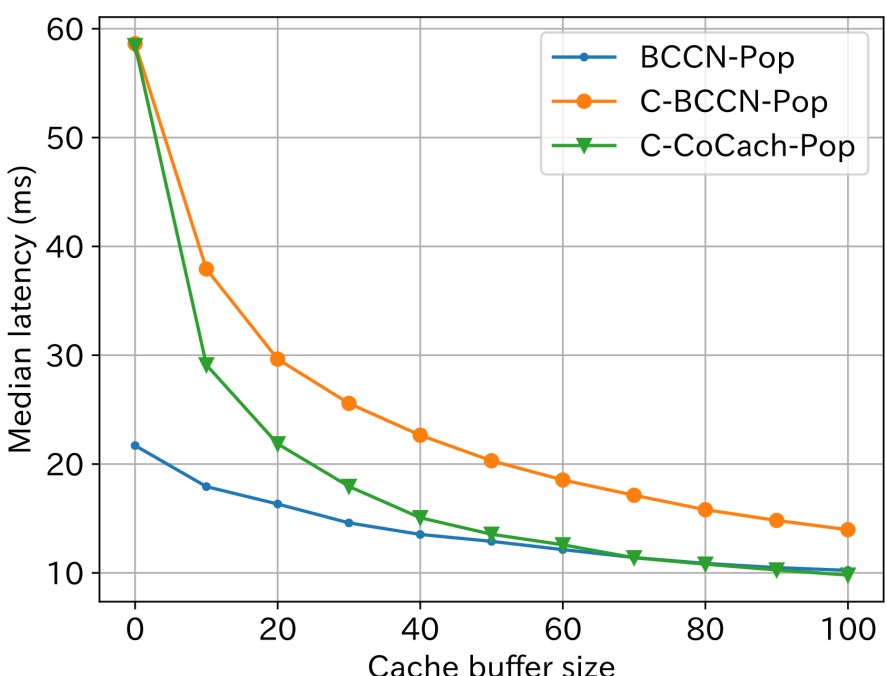

**Figure 20.** Median latency for retrieving the requested content with respect to cache buffer size (Scenario 2).

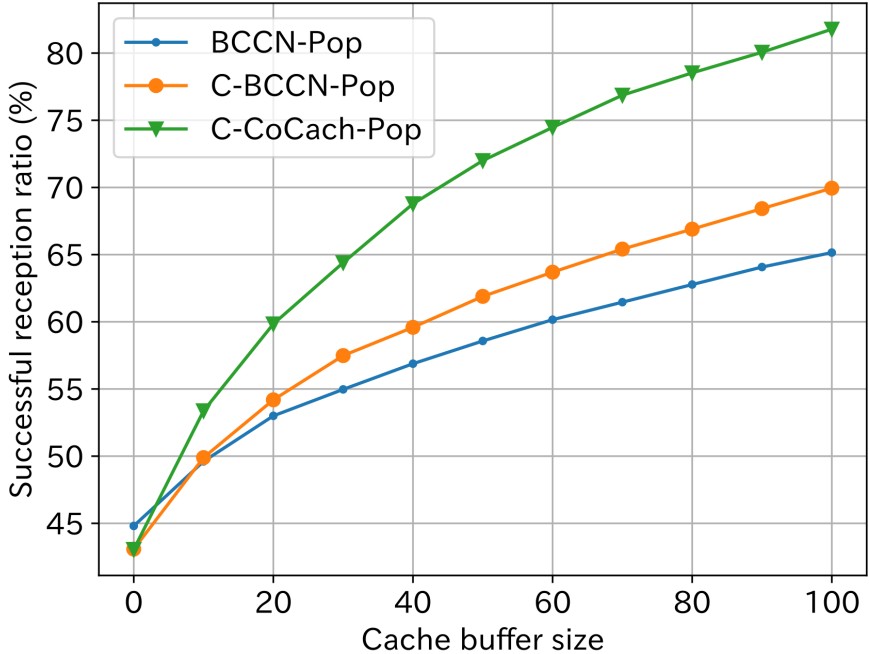

**Figure 21.** Success ratio of retrieving requested content with respect to cache buffer size (Scenario 2).

Figures 22 and 23 show the results for Scenario 3, for which the number of vehicles is 300. The proposed C-CoCach-Pop method still achieved better results than C-BCCN-Pop for each metric, although the differences get smaller than for the straight road in Scenarios 1 and 2 due to the obstruction of roadside buildings and complex vehicle mobility in the urban area.

Finally, we compared our method with an existing method: ECV+ [3]. Figures 24 and 25 compare our method and ECV+ for Scenario 2. Our method achieved better results because (1) ECV+'s probabilistic caching makes it difficult to fill vehicles' cache buffers, and (2) ECV+ does not consider content popularity.

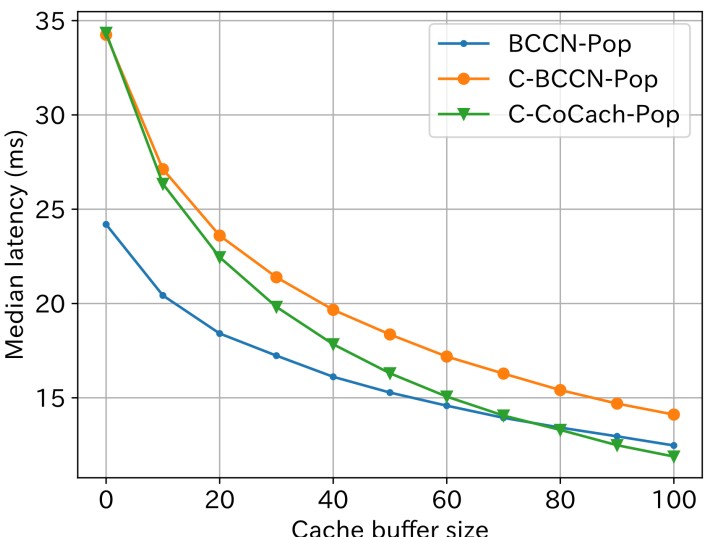

**Figure 22.** Median latency for retrieving the requested content with respect to cache buffer size (Scenario 3).

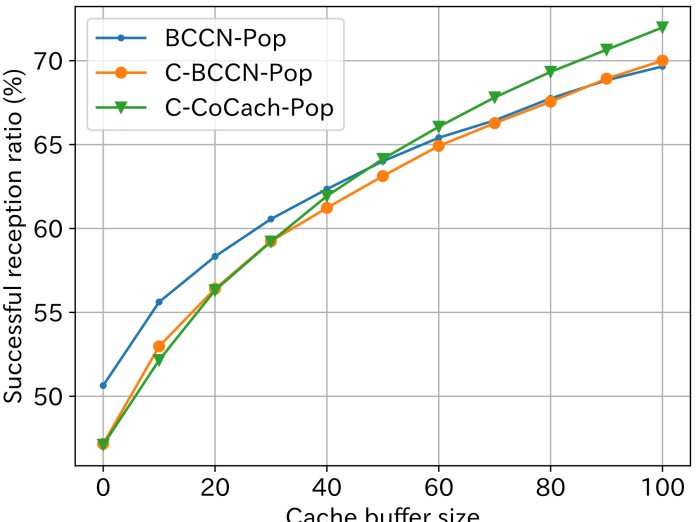

**Figure 23.** Success ratio of retrieving requested content with respect to cache buffer size (Scenario 3).

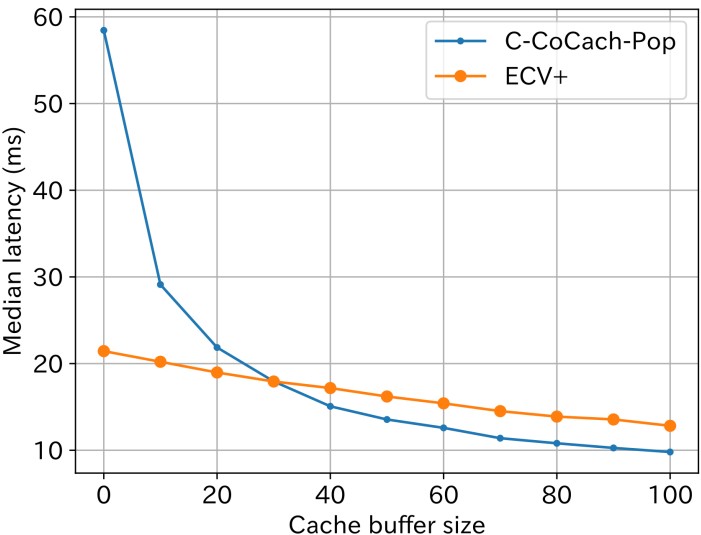

**Figure 24.** Median latency for retrieving the requested content with respect to cache buffer size (Scenario 2).

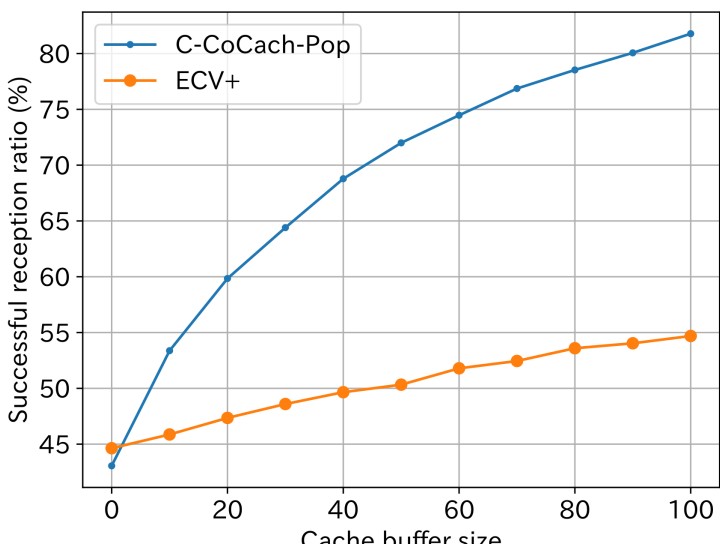

**Figure 25.** Success ratio of retrieving requested content with respect to cache buffer size (Scenario 2).

## 5. Conclusions

This paper has proposed a new collaborative caching method for fetching content efficiently in vehicular networks by extending our previous method [9] with popularity inference. By organizing vehicles in clusters, the proposed method has the following merits: (i) a CH learns all content cached in its cluster to reply to content requests, which improves the cache hit rate, (ii) a CH efficiently uses all cache buffer space in its cluster by avoiding duplicate caches and storing more diverse content, which increases the probability of locally responding to a content request. Further, we let vehicles infer content popularity and store popular content in their clusters, which increases the number of requests that could be completed within a cluster. In this way, the proposed method reduces the fetching latency of data packets by 33% and improves the packet reception rate by 17% compared to C-BCCN-Pop in an environment where the number of vehicles is 200. The proposed method is expected to enable rapid acquisition of road conditions.

In the future, we will further refine cache management to improve cache usage to get better results in both straight roads and urban models and will evaluate our method using actual V2V communications.

**Author Contributions:** Conceptualization, S.T.; methodology, H.T.; software, H.T.; validation, H.T.; writing—original draft preparation, H.T.; writing—review and editing, S.T.; visualization, H.T.; supervision, S.T. All authors have read and agreed to the published version of the manuscript.

**Funding:** This research received no external funding

**Data Availability Statement:** Data are contained within the article.

**Conflicts of Interest:** The authors declare no conflicts of interest.

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
