# Peer review of "Efficient V2V Communications by Clustering-Based Collaborative Caching†"

_electronics, doi:10.3390/electronics13050883_

Round 1

Reviewer 1 Report

Comments and Suggestions for Authors

The authors present in this paper a collaborative caching approach where vehicles are organized into clusters, and each cluster is overseen by a designated head responsible for managing caches among all vehicles within the cluster. That approach overcome the challenges of scenarios where vehicles are unable to utilize contents stored in nearby vehicles, beyond the communication path, and redundant caching of the same content occurred among nearby vehicles. Although the paper is written clearly, the reviewer has the below comments to be addressed before publishing it:

1.      The writing of the abstract must include clear details about the obtained results compared to the previous schemes exist in the literature.

2.      The contribution presented in this paper should be clarified at the end of the introduction with enough details. Especially, compared with the published part in the conference in [9].

3.      Sections I and II, which are the introduction and the related works, take a huge part from the paper. They can be summarized concentrating on the most important progress in the field.

4.      The resolution of some figures should be with higher resolution quality such as Figure 7.

5.      The results should be compared also with more works exist in the literature.

6.      Some updated and related references can be added to ease following the paper’s reading such as:

(1)    M. Belkheir, Z. Qasem, M. Bouziani, and A. Zerroug, "Ad Hoc Network Lifetime Enhancement by Energy Optimization," Ad Hoc Sens. Wirel. Networks, vol. 28, no. 1-2, pp. 83-95, 2015.

(2)    Z. A. Qasem, H. A. Leftah, H. Sun, J. Qi, J. Wang, and H. Esmaiel, "Deep learning-based code indexed modulation for autonomous underwater vehicles systems," Vehicular Communications, vol. 28, p. 100314, 2021.

Reviewer 2 Report

Comments and Suggestions for Authors

The author intends to design an effective cluster based cooperative caching V 2V communication method and provides content based network communication technology. Some of the content has reference value, but there are some issues in the paper.

1. The author did not provide valuable new methods and technologies, only a combination of existing small techniques, and did not demonstrate their universality.

2.The method designed by the author is not specific enough for other readers to reproduce, and specific implementation processes, formulas, and steps should be provided.

3. The author's V2V communication modeling should be more universal and valuable formulas or conclusions should be derived.

4.How do the key parameters of the paper work? Few mathematical formulas cannot derive the relationship between performance indicators and system parameters?

5. The source literature for the two comparison methods has not been provided, and their simulation conclusions are unreliable.

6. The author's contribution has been exaggerated, with the caching framework based on collaborative content being reflected in both references [13] and [15].

7. During simulation, the author should provide more comparison results of existing methods to demonstrate the superiority and effectiveness of the designed method.

8.During simulation, the author should clearly introduce the relationship between performance and system parameters.

9. There should be actual V2V communication systems and measured data to prove the use of the designed method.

10. It is best for the author to provide simulation comparisons of communication time for various methods.

Comments on the Quality of English Language

Some sentences can be expressed more accurately. 

Reviewer 3 Report

Comments and Suggestions for Authors

1. The abstract of the paper states that "Extensive simulation results demonstrated that this innovative approach effectively reduced content delivery latency and enhanced the success rate of retrieving requested content" but it is unclear how extensive the simulation was. 

2. The abstract should contain a summary of the statistical results, the conclusion, and the future directions of the work.

3. Referencing should be ordered by the authors in accordance with the citations in the text, not the other way around.

4. In their article, they describe it as "an effective solution to address (1) and (2). [6] advocates" what the authors want to highlight?

5. What the authors want to say in the introduction line 47 to 56, please make it a literature review since that is what they want to say.

6. There is not enough literature review in the paper.

7. The authors should provide the problem before the contributions in the introduction.

8. Figure 1 is not a research figure. The author needs to make it a more researchable figure.

9. What is the use of Figure 7?

10. The results are encouraging, but it is not mentioned how the authors got these results and what was the problem.

11. The references need to be updated.

Comments on the Quality of English Language

Extensive editing of English language required

Reviewer 4 Report

Comments and Suggestions for Authors

In this paper, a distributed content caching mechanism had been proposed for V2V networks which improves on existing techniques by considering the popularity of the content data and managing its distribution within and between different clusters of vehicles. The performance of the proposed technique had been evaluated and compared against other techniques that do not consider collaborative caching and/or content popularity. The results from the simulation had demonstrated the gains in performance and mainly in terms of latency and successful reception. These results demonstrate the increased efficiency of the proposed algorithm compared to the existing methods. 

Overall, the merits of the proposed solution have been described and demonstrated quite well in the article. While the proposed collaborative caching algorithm had been described in detail in Section 3, it was slightly challenging to follow which may be due to the order in which the different parts of the algorithm had been described. However, the examples were helpful at clarifying the various functions of the algorithm. It would be worthwhile to include an algorithm (or a flowchart) that can help to better define the different functions of the proposed mechanism and the relation between them. Nevertheless, the work described in this paper is good and I believe would be a significant contribution to the field of research on content caching in V2V networks.

Reviewer 5 Report

Comments and Suggestions for Authors

The research presented a caching strategy in V2V communications by using CCN techniques to reduce latence and storage. Here are some comments from reviewer. 

1) The authors should consider and evaluate the overhead, since all vehicles are equiped with a CM table.

2)  What happens if an OV receives a beacon from another OV?

3) What types of comfortability data or services the authors considered/selected when building the system model and conducting simulations? Why you select this?

4) The data packet is 512 bytes long, it is too long for short messages and too short for many content services, such as video streaming. Why the authors choose it?

5) The text, including the legends and axis titles in the figures are too small.

6) The reviewer suggest the authors considering and discussing the following research:

  C. Wu, T. Yoshinaga, Y. Ji, T. Murase and Y. Zhang, "A Reinforcement Learning-Based Data Storage Scheme for Vehicular Ad Hoc Networks," in IEEE Transactions on Vehicular Technology, vol. 66, no. 7, pp. 6336-6348, July 2017, doi: 10.1109/TVT.2016.2643665.

7) Do the author consider congestion control in CCN network like in the following paper? 

H. Bai, H. Li, J. Que, M. Zhang and P. H. J. Chong, "DSCCP: A Differentiated Service-based Congestion Control Protocol for Information-Centric Networking," 2022 IEEE Wireless Communications and Networking Conference (WCNC), Austin, TX, USA, 2022, pp. 1641-1646, doi: 10.1109/WCNC51071.2022.9771825.

8) The reviewer suggest the authors dicussing and comparing clustering in the following paper. 

M. Zhang, Y. Dou, V. Marojevic, P. H. J. Chong and H. C. B. Chan, "FAQ: A Fuzzy-Logic-Assisted Q-Learning Model for Resource Allocation in 6G V2X," in IEEE Internet of Things Journal, vol. 11, no. 2, pp. 2472-2489, 15 Jan.15, 2024, doi: 10.1109/JIOT.2023.3294279.

Comments on the Quality of English Language

N/A

Reviewer 6 Report

Comments and Suggestions for Authors

1.  The authors propose a collaborative caching method, in which vehicles are grouped into clusters and each cluster has a designated head responsible for managing caches across all vehicles within the cluster. In this way, this method enables vehicles to use the contents cached at adjacent vehicles that are not directly on a communication path.

2.       In the figure 1, the overview of the proposed method should be demonstrated in detail.

3.  In the figure 4, communication process in the proposed method should be demonstrated in detail.

4.       In the figure 5, request counter table should be demonstrated in detail.

5.       Please thoroughly revise the language before your submission.

Comments on the Quality of English Language

Minor editing of English language required.

Round 2

Reviewer 1 Report

Comments and Suggestions for Authors

the authors addressed my raised comments. I think the paper is ready to be published.

Reviewer 2 Report

Comments and Suggestions for Authors

The authors have carefully revised the paper according to my suggestions, and now it can be published. 

Reviewer 3 Report

Comments and Suggestions for Authors

1. The authors has response of my previous comments only the cited references are in order like {1,4,5], [1, 7,8] and so on.

Comments on the Quality of English Language

Minor editing of English language required

Reviewer 5 Report

Comments and Suggestions for Authors

The authors dealt with all my concerns from last round of round.